# Analysis of TRIP Steel HCT690 Deformation Behaviour for Prediction of the Deformation Process and Spring-Back of the Material via Numerical Simulation

**DOI:** 10.3390/ma17030535

**Published:** 2024-01-23

**Authors:** David Koreček, Pavel Solfronk, Jiří Sobotka

**Affiliations:** Department of Engineering Technology, Faculty of Mechanical Engineering, Technical University of Liberec, Studentská 1402/2, 46117 Liberec, Czech Republic; pavel.solfronk@tul.cz (P.S.); jiri.sobotka@tul.cz (J.S.)

**Keywords:** high-strength steel, TRIP steel, yield criterion, hardening law, spring-back, sheet metal forming, FEM, numerical simulation, PAM-STAMP 2G

## Abstract

This paper deals with the analysis of TRIP steel HCT690 deformation behaviour. The mechanical properties and deformation characteristics of the tested material are determined using selected material tests and tests that consider the required stress states used to define the yield criterion boundary condition and subsequent deformation behaviour in the region of severe plastic deformation. The measured data are subsequently implemented in the numerical simulation of sheet metal forming, where they are used as input data for the computational process in the form of a selected material model defining the yield criterion boundary and, furthermore, the material hardening law during deformation of the material. The chosen numerical simulation process corresponds to the sheet metal forming process, including the subsequent spring-back of the material, when the force does not affect the material. Furthermore, the influence of the chosen computational model and selected process parameters on the deformation and spring-back process of the material is evaluated. In addition to that, at the end of the paper, the results from the numerical simulation are compared with experimentally produced sheet stamping.

## 1. Introduction

Today, high demands are placed on sheet metal stampings, especially in terms of strength, surface quality, dimensional accuracy and stability. The stiffness and strength of a given stamping are, to a large extent, influenced by the shape concept of such a part, the material used for its production and, of course, the technological process of production. A necessary condition for the proper design of a suitable technological production process is therefore, first of all, to calculate the deformation process of the selected material to achieve the final shape of the stamping with the required quality. This is also related to the permissible thinning of the material, sufficient deformation of the sheet, wrinkling of the material, elimination of visual defects, etc.

The difficulty in achieving the shape and dimensional accuracy of the formed part is also closely related to the material spring-back, which results from the portion of elastic deformation accompanying the deformation process of the material. This undesirable phenomenon of material spring-back during the forming process can be mostly eliminated just by the appropriate design of technological operations and the shape correction of the forming tools. Contemporary trends give the green light to the ever-evolving design changes in new types of car bodies, which significantly increase the requirements for the shape and dimensional accuracy of formed car-body parts, making it necessary to introduce new technologies and methods to achieve these manufacturer-driven requirements.

This branch includes, in particular, an increasing portion of the mathematical modelling of technological processes already in the pre-production and production stages as well. This provides us with the possibility to react quickly, variably, in time and, above all, economically to current issues during the conception and production of formed parts. These aspects are the reason for and directly encourage research in the field of introducing new types of materials: the description of their deformation behaviour and the refinement of the material numerical models used in the mathematical modelling of these processes.

In the case of drawing technology, the state of stress on the drawing edge is repeatedly changed when the formed material is first bent and then straightened again on the drawing edge. Thus, the so-called Bauschinger effect occurs during material hardening, and this phenomenon to a large extent complicates the proper definition and mathematical description of the material hardening during deformation in the numerical simulation of the forming process. Moreover, it influences and is closely related to the relevant yield criterion used in the numerical simulation of the deformation process. This issue is currently very widely discussed and can be found, e.g., in the publications of Henk Vegter from Tata Steel Europe Limited TATA [1,2,3,4,5,6,7] or Pavel Hora from ETH Zürich [8,9,10,11,12,13]. In addition, this topic has also been discussed, e.g., by Takeshi Uemoriho from Okayama University [14,15,16,17,18,19], Takeshi Yoshida from Hiroshima University [18,20,21,22,23,24,25], etc.

A fundamental problem in the numerical modelling of the elastic, elastic–plastic and plastic behaviour of a material is the consideration and definition of its anisotropic behaviour as well as the description of its hardening during plastic transformation [26]. The anisotropic behaviour of a material affects the position of the yield criterion boundary condition and different loading paths need to be considered with respect to the formed material’s stress state [27]. The studies of Prof. H. Vegter and others [2,7,9,27] demonstrate the advantages of using modern yield criteria that take into account the anisotropy of the material as well as the influence of different stress states on the formed material. Furthermore, it has been shown (with respect to the Bauschinger effect, which is described above) that utilisation of the isotropic hardening law is insufficient and that the assumption of non-isotropic hardening of the material must be taken into account if accurate results are required from the material model [26]. In order to capture and describe the effect of yield strength variation under alternating (tension–compression) loading, the kinematic hardening law has been proposed, as described, for example, by Yoshida-Uemori [28]. In the case of experimental measurements of cyclic plasticity, only a limited number of papers have been published, one of the reasons for this being that motivation in the field of cyclic plasticity research has only arisen in the area of sheet metal forming numerical simulations [28] and there is considerable complexity in performing cyclic tests to obtain stress–strain characteristics under alternating loading (tension–compression) [28]. This problem is addressed, for example, in a publication by Prof. Yoshida et al. [29], where the authors deal with the deformation behaviour of high-strength steel under cyclic loading and the design of a suitable cyclic testing methodology for the material [29]. Cyclic loading can be viewed as a specific case of changing the loading path and considering the Bauschinger effect as an effect associated with the reversal of this loading path [26]. Indeed, in the general case, the initial deformation does not only affect the size and position of the yield strength, but also its shape [26]. This issue is the subject of research in publications such as Barlat et al. (2011) [30], Barlat et al. (2013) [31], Feigenbaum et al. (2012) [32] and Freund et al. (2012) [33].

From published papers, it is possible to make conclusions and assumptions that for the definition and description of the advanced mathematical modelling of material behaviour during deformation, and its subsequent spring-back, it is necessary to use modern yield criteria with the influence of the material anisotropy to include the influence of the Bauschinger effect. However, for its proper definition, it is necessary to perform and implement cyclic tests with fully reversed cyclic loading.

## 2. Materials and Methods

In the following sections, research, characterisation and description of the tested material, material tests necessary to describe the deformation behaviour of the material and the issues about the numerical simulation of sheet metal forming are presented.

### 2.1. TRIP Steel HCT690 (EN 10346)

Transformation-induced plasticity steel (TRIP steel) is a material whose structure is based on a ferritic–bainitic matrix. Inside this matrix, there is 5 to 10 percent of the retained metastable austenite in the initial state, which is transformed into a martensitic phase just during the subsequent deformation process (the so-called “TRIP effect”). In order to achieve this transformation, sufficient content of carbon must be present in the retained austenite to decrease the martensite start temperature to room temperature [34,35].

The hardening mechanism during deformation is similar to that of dual-phase (DP) steels, where dislocations are accumulated around the grain boundaries of the martensitic phase within the basic ferritic–bainitic matrix. In TRIP steels, this phenomenon is additionally accompanied by a TRIP effect, whereby retained austenite gradually transforms into martensite as the material is increasingly loaded. This increases the hardening rate at higher strain rates. The hardening rate in these steels is significantly higher than in conventional HSS steels. Moreover, a high hardening rate is maintained under higher strain rates, thus giving an advantage over DP steels. The structure of a TRIP steel during the material-forming process, where the deformation process transforms the retained austenite involved in the deformation process into martensite, is graphically shown in Figure 1 [35,36].

The magnitude and rate of the TRIP effect are influenced by the chemical composition of the material. The level of transformation at which the retained (residual) austenite begins to transform into martensite is controlled by the carbon content. At lower levels of carbon content, the retained austenite begins to transform into martensite almost immediately after deformation, which increases the hardening rate and the formability of the material during the forming process. At higher carbon contents, the retained austenite is more stabilised and begins to transform only at transformation levels above those generated during forming. At these carbon levels, the retained austenite remains within the structure and transforms into martensite only during subsequent deformation [35,36].

A very important condition for the presence of retained austenite in the material structure is also the choice of appropriate heat or thermo-mechanical treatment. The production process of these steels, similar to that of dual-phase steels, consists of controlled cooling of the austenitic or ferritic–austenitic structure in combination with rolling of the material. In the case of hot-rolled sheets, the rolling is followed by a ferritic change after the ferrite start has been achieved. This transformation continues until a dual-phase ferritic–austenitic structure is achieved. In cold-rolled sheets, the formation of the ferritic–austenitic structure is followed by heating the material and subsequent annealing at temperatures between Ac1 and Ac3. In both cases, this is followed by cooling to the bainitic start temperature, followed by holding at this temperature to create the bainitic phase. The transformation can continue long enough to produce the desired ratio of the different phases (about 15% of retained austenite, 60% ferrite and 25% bainite), followed by pre-cooling to the ambient temperature. A schematic representation of the TRIP steel production by heat treatment combined with hot or cold rolling of material is graphically illustrated in Figure 2 [37,38].

Steels with transformation-induced plasticity (TRIP steels) contain a larger carbon content than dual-phase steels; the carbon content here is approximately 0.2%. Sufficient carbon content is important to stabilise the retained austenitic phase at ambient temperature. In addition, there is about 1.5–2% austenite forming manganese, which also suppresses the pearlitic transformation. A higher silicon content (about 1.8%), which is sometimes substituted by aluminium, supports the formation of ferrite. Both of these elements help to maintain the required carbon content in the retained austenite and suppress the carbide formation (cementite formation) during the bainitic transformation. This appears to be essential for the production of TRIP steels. Furthermore, small amounts of accompanying elements such as phosphorus, sulphur, copper or nickel may be present [36,37,40,41,42].

These steels have a very wide range of applications as they can be produced or adapted to reveal excellent formability for the manufacture of complex parts or to exhibit high material strength during subsequent deformation (for example, during traffic accidents), which is allowed by the high amount of absorbed energy. These properties predetermine the use of TRIP steels for a variety of automotive car-body parts or as safety components in the deformation zones—as illustrated in Figure 3. The production batch of these steels their use, e.g., TRIP 350/600 (pillar reinforcements), TRIP 400/700 (side rails, bumpers), TRIP 450/800 (side panels, roof rails), TRIP 600/980 (upper B-pillars, roof beams, engine mounts, front and rear beams, seat frames), TRIP 750/980 and so on [36].

### 2.2. Methodology of Calculation the Sheet-Metal-Forming Numerical Simulation in PAM-STAMP 2G Software

The main tool for the calculation of numerical simulations in the software PAM-STAMP 2G v2015 are the numerical methods of non-rigid body mechanics. These methods use the finite element method (FEM) for the calculation. The finite element method is currently classified as the most accurate tool to be used in numerical simulations of the forming process.

The FEM, unlike the classical variational methods, is based on the approximating the course of a given variable. The finite element method constructs and prescribes the resulting function using non-zero approximations only in limited volumes—finite elements. These elements are generated by subdividing the region of interest into geometrically simple, mutually disjointed subregions. Planar regions are usually converted into a triangular or polygonal elements, volume regions are converted into tetrahedra, cubes, etc. Such conversion is referred to as a meshing, which creates a so-called finite element mesh. Depending on how many elements are contained in this mesh, just enough local approximations can be found to “spline” model the function of interest. The functions to approximate are generally chosen to be quite simple, most often with just polynomial dependence, where the number of arguments depends on the type of problem. Beyond the boundaries of the individual element, functions are defined by zero. At the common element boundaries, the continuity requirement of the function must be satisfied, which restricts the dependence of the elementary functions’ combination coefficients in the individual approximation prescriptions of the adjacent elements. This problem is solved by eliminating the individual coefficients via function values at properly chosen points of the elements, most commonly referred to as nodes (contact points). These nodes are preferably placed at the boundaries of elements, especially at their vertices [43,44].

In the numerical simulation, all components involved in the forming process are converted to a computational mesh—the so-called meshing process. For a non-deformable tool, the mesh represents only its geometry and the finite elements are only taken as facets that are used to describe the contact in the relevant surfaces. Conversely, in the case of deformable process components (e.g., blank or tube), the finite elements of the mesh are just small “pieces” of material, whose behaviour is already prescribed. The mechanical phenomena that occur in the material during its deformation are reproduced by a large number of these elements. The finer the mesh of these elements, the more accurate the computation of the forming process. On the other hand, as the mesh becomes finer, the computational time increases. A 2-node element (bar), a 3-node element (triangle), a 4-node element (quadrangle), or an element of 6 or 8 nodes can be used as the final element. Each node has two types of degrees of freedom—translation and rotation [45,46].

Depending on the type of numerical simulation computation (implicit or explicit algorithm), the computation consists of increments or time steps. At each node, depending on the type of computation algorithm, the position, velocity, acceleration and force are detected. During the numerical simulation of the metal-forming process, these values are permanently calculated at each node and, from them, magnitude of the actual stress and strain is subsequently calculated. This algorithm is repeated in all elements throughout the simulation process. Boundary conditions of the given process are used to remove degrees of freedom (so-called locking). In order to precisely describe the deformation process, the simulation must be supplemented with the material characteristics and dimensions of the deformed blank [45,46].

### 2.3. Selected Material Computational Models of Numerical Simulation in PAM-STAMP 2G Software

In the numerical simulation, different yield criteria are generally available for the definition of material models. These yield criteria can work with an isotropic or anisotropic material condition. Quite a lot of different approaches can be used in the sheet metal numerical simulations to define the relevant yield criterion, either isotropic or anisotropic. On the one hand, some conditions solve the computation of the material’s anisotropic behaviour during its deformation via a mathematical approach with numerical equations (for example, Hill 48, Barlat, etc.). On the other hand, some conditions describe the anisotropic behaviour of the material during deformation by externally determined material characteristics and input quantities, which are obtained through a greater or lesser number of specific experimental material tests (Vegter model, Yoshida, etc.) [46].

As a result of the reality that this is a sheet-metal-forming technology for thin sheets, the stress in the thickness direction σ_3_ is neglected. Thus, the yield criterion takes a planar expression and can be defined by an ellipse (see Figure 4), which forms the yield criterion boundary within the plane of the main stresses σ_1_ and σ_2_. The shape of the ellipse, which represents the relevant yield criterion, can be controlled by the mathematical expressions used to calculate the anisotropy or by the external material characteristics obtained from the experimental material testing. In order to compile the yield criterion using the experimental inputs to describe the material anisotropy, so-called reference (control) points of the ellipse are needed. These points represent the individual mechanical tests of the material. For the common yield criteria (e.g., Hill 48), it is sufficient to perform only a static tensile test. However, to define advanced materials models that work with more accurate yield criteria (e.g., Vegter yield criterion), more input material characteristics are required, which leads to the need to carry out additional material tests such as biaxial tests, compression tests, shear tests, etc. (see Figure 4) [2,3,46].

The individual material models used in the numerical simulation depend on the chosen yield criterion but also on the material hardening law as the material deforms. Therefore, in the computational model of the sheet metal forming numerical simulation, the yield criterion itself is supplemented by a material hardening law during deformation. The first possibility represents the isotropic hardening law, which is defined by Krupkowski approximation of the stress–strain curve along the reference direction 0° with respect to the rolling direction. Krupkowski approximation of the hardening curve is based on Equation (1) and is illustrated in Figure 5 [2,3,46].
(1)$$\sigma ={C\;\cdot \;\left(\varphi +\varphi_{0}\right)}^{n}$$
where:*C*—strength coefficient (MPa),*n*—strain hardening exponent (-),*φ*_0_—offset true strain (-).

**Figure 5 materials-17-00535-f005:**
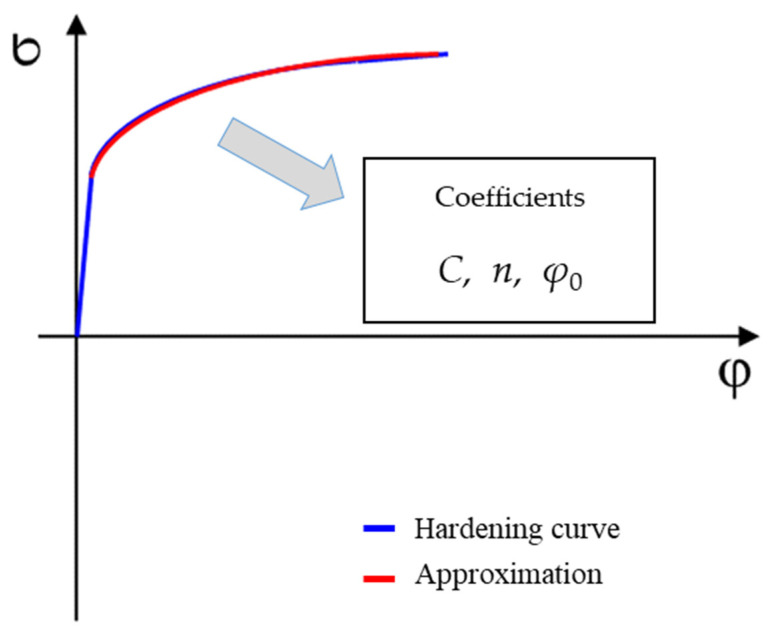
Illustration of the stress–strain curve approximation for isotropic hardening law definition.

The second more advanced material model is the Yoshida kinematic hardening law. This model also takes into account the so-called Bauschinger effect, which describes the change in the yield strength of material due to alternating cyclic loading (tensile–compression). This phenomenon is commonly encountered in the real forming processes, e.g., during bending and subsequent straightening of the formed material on the drawing edge, etc. This model is defined by hysteresis loops obtained from cyclic testing under fully reversed (tensile–compression) cyclic loading. The advantage of the kinematic hardening law compared to the isotropic hardening law is illustrated in Figure 6a [46].

As already described above in the general yield criteria, the Bauschinger effect due to changing the material loading shifts the boundary surface, which is illustrated by the Yoshida kinematic hardening law in Figure 6b. This hardening law is generally described by the equations below [46].

The yield function at the initial state *f_0_* can be defined by Equation (2), where function Φ denotes the yield surface and *Y* is its boundary [46].
(2)$$f_{0}=\;\;\phi \;\left(\sigma \right)-Y=0$$

This function can be further rearranged into Equation (3), where *σ* denotes a Cauchy stress and *α* means the backstress. The associated flow rule is subsequently written as Equation (4). The bounding surface is expressed by Equation (5), where *β* is the centre of the bounding surface and *B* and *R* are its initial size and isotropic hardening component. The relative kinematic motion of the yield surface regarding the bounding surface is expressed by Equation (6). The evolution equation for this motion is given by Equation (7). $\dot{\bar{\varepsilon }}$ denotes the effective plastic strain rate—see Equation (8), which is defined via the second invariant *D*^p^. Finally, *C* represents a material parameter that controls the rate of kinematic hardening [46].
(3)$$f=\;\;\phi \;\left(\sigma -\alpha \right)-Y=0$$
(4)$$D^{p}=\;\;\frac{\partial f}{\partial \sigma }\;\dot{\lambda }$$
(5)$$F=\varphi \left(\sigma -\beta \right)-\left(B+R\right)=0$$
(6)$$\dot{\alpha_{*}}=\dot{\alpha }-\dot{\beta }$$
(7)$$\dot{\alpha_{*}}=C\left[\frac{a}{Y}\left(\sigma -\alpha \right)-\sqrt{\frac{a}{\bar{\alpha_{*}}}}\;\alpha_{*}\right]\dot{\bar{\varepsilon }}$$
(8)$$\dot{\bar{\varepsilon }}=\sqrt{\frac{2}{\bar{3}}D^{p}:\;D^{p}}\;,\;\;\bar{\sigma_{*}}=\Phi \;\left(\alpha_{*}\right),\;a=B+R-Y$$

To describe the global work hardening for the bounding surface, the following evolution of Equation (9) is used, where *R*_sat_ is the saturated value of the isotropic hardening stress at infinitely large plastic strain and *m* is a material parameter that controls the rate of isotropic hardening [46].
(9)$$\dot{R}=m\;(R_{sat}-R)\dot{\bar{\varepsilon }}\;$$

The kinematic hardening and non-isotropic hardening region during stress reversals are assumed for the bounding surface to describe the permanent softening and work-hardening stagnation during stress reveals. For the kinematic hardening of the bounding surface, the evolution of Equation (10) of the Armstrong–Frederick type was used, where *β*′ and $\dot{\beta^{\prime }}$ are the deviatoric components of *β* and its objective rate, *b*_sat_ is a material parameter [46].
(10)$$\dot{\beta^{\prime }}=m\;(\frac{2}{3}{\;b}_{sat}\;D^{p}-\beta^{\prime }\dot{\bar{\varepsilon }}\;)\;$$

The so-called non-isotropic hardening (non-IH) of the bounding surface at a certain range of reverse deformation is used to describe the work hardening stagnation. That is why the non-IH surface in the deviatoric stress space is defined [46].

From some experimentally obtained data about the stress–strain curves, it arises that the region of work hardening stagnation increases with the accumulated plastic deformation. Because of that, the kinematic motion of the non-IH surface centre in the direction defined in Equation (11) is assumed. From the consistency condition, it is valid that the centre point of the bounding surface should be either on or inside of surface *g*_σ_ (see Figure 7) [46].
(11)$$\dot{q^{\prime }}=\mu \;(\beta^{\prime }-q^{\prime })$$

Moreover, in Equation (12) the presumption expressed in Equation (13) is assumed, where *h* is a material parameter that determines the rate of surface expansion *g*_σ_. Then, Equation (14) must be valid [46].
(12)$$\;\mu =\frac{3\;\left(\beta^{\prime }-q^{\prime }\;\right):\;\dot{\beta^{\prime }}}{2r^{2}}-\frac{\dot{r}}{r}\;$$
(13)$$\;\;\dot{r}=h\Gamma ,\;\;\Gamma =\frac{3\;\left(\beta^{\prime }-q^{\prime }\;\right):\;\dot{\beta^{\prime }}}{2r}\;$$
(14)$$\;\mu =\frac{\left(1-h\right)\;\Gamma }{r}\;\;$$

A larger value of *h* means a rapid expansion of the non-IH surface, which leads to the prediction of smaller cyclic hardening. Since work hardening stagnation appears during the material hardening, the initial value of parameter *r* may be assumed to be very small, i.e., *r = r*_0_ [46].

The elastic–plastic constitutive equation is expressed by Equation (15), where C denotes the tensor of elastic modulus and subsequently *H*_kin_ is the rate of kinematic hardening as shown in Equation (16) [46].
(15)$$\;\dot{\sigma }=\left[C-\frac{C:\frac{\partial f}{\partial \sigma }\;\times \frac{\partial f}{\partial \sigma }:C}{\sqrt{\frac{2}{\bar{3}}}\;H_{kin\;}|\left|\frac{\partial f}{\partial \sigma }\right||+\frac{\partial f}{\partial \sigma }:C:\frac{\partial f}{\partial \sigma }}\right]\;$$
(16)$$\;H_{kin\;}=\left[\frac{C\;a+m\;b_{sat}}{Y}\left(\sigma -\alpha \right)-\left(C\;\sqrt{\frac{a}{\bar{\alpha_{*}}}}\;\alpha_{*}+m\;\beta \;\right)\right]:\;\frac{\partial f}{\partial \sigma }\;$$

In this model, the size of the yield surface is already kept constant. However, the stress–strain response during unloading after plastic deformation is no longer linear but slightly curved due to the Bauschinger effect. To describe this phenomenon, the following Equation (17) of plastic-strain-dependent Young’s modulus is used [46].
(17)$$E=E_{0}-(E_{0}-E_{a})\left[1-exp(-\xi \varepsilon )\right]\;$$

In Equation (17), *E*_0_ and *E*_a_ stand for the Young’s modulus for virgin (original) and infinitely large pre-strained materials, respectively [46].

The following subsections describe both Vegter yield criteria and material computation models for the sheet metal numerical simulation process, ordered by the complexity of input data determination for their definition. Both the Vegter “Standard” and Vegter Lite yield criteria work with an anisotropic yield condition. The principal stresses *σ*_1_, *σ*_2_ and angle of the coordinate system rotation *Φ* in the Vegter yield criterion are defined using the following Equations (18)–(20) [46].
(18)$$\sigma_{1}=\;\;\frac{\sigma_{xx}+\sigma_{yy}}{2}+\sqrt{{\left(\frac{\sigma_{xx}-\sigma_{yy}}{2}\right)}^{2}+{\sigma_{xy}}^{2}}\;$$
(19)$$\;\;\sigma_{2}=\;\;\frac{\sigma_{xx}+\sigma_{yy}}{2}-\sqrt{{\left(\frac{\sigma_{xx}-\sigma_{yy}}{2}\right)}^{2}+{\sigma_{xy}}^{2}}\;$$
(20)$$cos(2ϴ)=\;\;\frac{\frac{\sigma_{xx}-\sigma_{yy}}{2}}{\sqrt{{\left(\frac{\sigma_{xx}-\sigma_{yy}}{2}\right)}^{2}+{\sigma_{xy}}^{2}}}\;\;$$
where:*σ*_1_—principal stress (direction 1) (MPa),*σ*_2_—principal stress (direction 2) (MPa),*σ*_xx_—stress in the direction 0° (MPa),*σ*_yy_—stress in the direction 90° (MPa),*σ*_xy_—shear stress (MPa),*ϴ*—angle of the coordination system rotation (°).

#### 2.3.1. Vegter Lite Yield Criterion

This yield criterion takes into account the direction-dependent anisotropy of the material. In order to define this model and establish the yield criterion, it is necessary to determine material constants, which can be obtained either experimentally or from the material data sheet of the tested material. In addition to that, relevant material data and characteristics are determined by relevant mechanical tests of the material [46].

The following material constants are needed to define the computational model:Young’s modulus *E*;Poisson’s ratio *μ*;Density ρ.

Furthermore, to properly define the Vegter Lite yield criterion, it is necessary to determine following material characteristics of the tested material (see Figure 8), which are determined experimentally by the mechanical testing of the material:Static tensile test;Hydraulic bulge test.

**Figure 8 materials-17-00535-f008:**
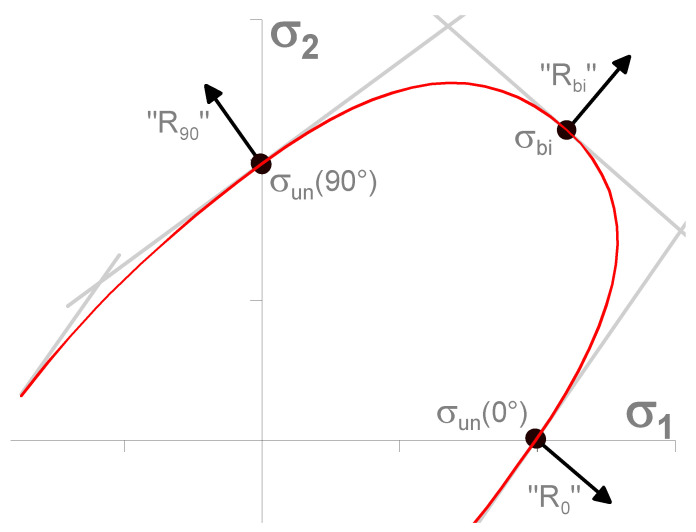
Vegter Lite Yield Criterion [3].

The last part for the complete definition of the material computational model in the numerical simulation is to describe the material hardening law during deformation. As described above, the material hardening can be described using an approximation of the average stress–strain curve (hardening curve) determined from a tensile test, or using characteristics obtained from a cyclic test under a fully reversed alternating cycle. By using these determined material characteristics, it is possible to define:Isotropic hardening law;(Yoshida) Kinematic hardening law.

#### 2.3.2. Vegter Yield Criterion

The Vegter yield criterion, like the Vegter Lite model, takes into account the direction-dependent anisotropy of the material. For its definition and determination of the yield criterion boundaries, more material data and characteristics are required. However, these data add hinge points to the ellipse that represent the yield criterion. On the other hand, they refine the computation process of numerical simulation [46].

The following material constants are needed to define the computational model:Young’s modulus *E*;Poisson’s ratio *μ*;Density ρ.

Furthermore, to properly define the Vegter yield criterion, it is necessary to determine selected material characteristics of the tested material (see Figure 9), which are determined by the following mechanical tests of the material:Static tensile test;Hydraulic bulge test;Plane strain tensile test;Shear test.

**Figure 9 materials-17-00535-f009:**
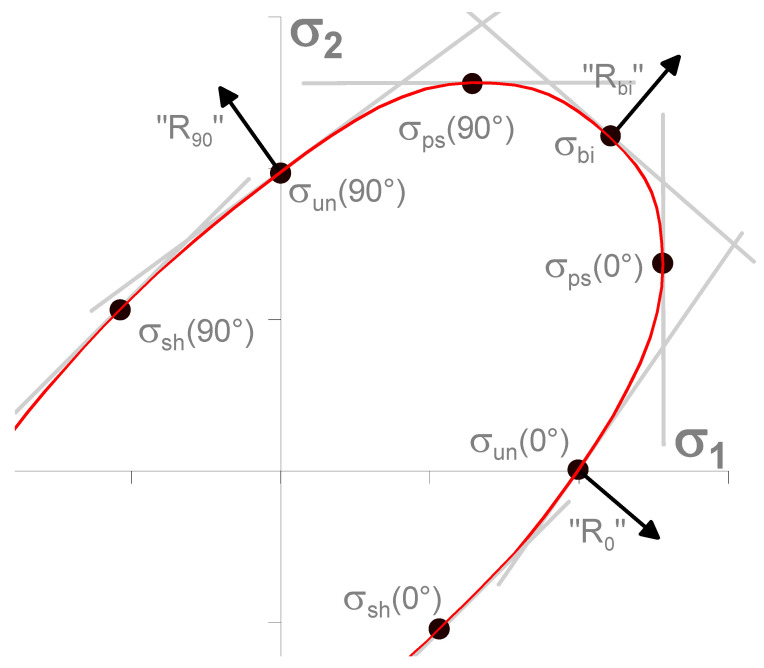
Vegter Yield Criterion [3].

As in the previous case (see Section 2.3.1), the last part for the complete definition of the material calculation model in the numerical simulation is to describe the material hardening law during deformation. As described above, it is possible to again define:Isotropic hardening law;(Yoshida) Kinematic hardening law.

## 3. Experimental Part

Concerning the material calculation models described above, material tests were used which can define both the yield criterion boundary of the selected material model and subsequently also the material hardening law during its deformation. The following subsections describe the individual material tests that were performed to obtain the needed material characteristics. These data were subsequently implemented in the numerical simulation via the software PAM-STAMP 2G.

### 3.1. Static Tensile Test

The static tensile test was used to determine the basic mechanical properties of the tested material. Values of the yield strength *R_e_* (even. proof yield strength *R_p_*_0.2_) and ultimate tensile strength *R_m_* as well as the uniform ductility A_g_ and total ductility *A*_80mm_ and Young’s modulus E were determined. In order to take into account the anisotropy of the material, the test must be carried out with specimens taken in the directions 0°, 45° and 90° with respect to the rolling direction of the material. The test was carried out in a standard way according to EN ISO 6892-1 [47]. A schematic diagram of the specimen loading during the static tensile test is shown in Figure 10. The dimensions of the testing samples were as follows: *L*_0_ = 80 mm, *w*_0_ = 20 mm and *t*_0_ = 1 mm.

The test was carried out using a TIRA Test 2300 testing device equipped with an integrated MFN-A-4-500 axial extensometer. In this case, the testing specimen was clamped with hydraulic jaws and loaded by force until failure. During the test, the force was determined by the load cell and the absolute elongation of the specimen was measured using the aforementioned axial extensometer. The evaluation of the basic quantities obtained from this test was carried out using the associated LabNET software v4.49.8944. Realisation of the static tensile test is illustrated in Figure 11.

### 3.2. Hydraulic Bulge Test

The hydraulic bulge test (HBT), or the material loading by the equi-biaxial stretching using hydraulic pressure, was performed to confirm that the stress state was the equi-biaxial stretching in the material. This test is used to obtain the basic mechanical properties of the material under equi-biaxial loading conditions. Here, the test specimen is loaded by hydraulic pressure until failure. A schematic diagram of the specimen during loading is shown in Figure 12.

This test was carried out on a CBA 300/63 hydraulic press using a jig designed to ensure equi-biaxial stretching of the testing specimen by hydraulic pressure, where the specimen is clamped between the blank-holders (more precisely a drawing die and blank-holder). Subsequently, the specimen is loaded by hydraulic pressure exerted by an external hydraulic unit. The principle of this test is schematically shown in Figure 13.

The hydraulic pressure (by hydraulic oil) was measured by a pressure sensor integrated in the jig and deformation of the material was determined using stereophotogrammetry (two scanning cameras). A contact-less MERCURY RT optical system from the company Sobriety was used in this case. The realisation of the hydraulic bulge test on a hydraulic press is illustrated in Figure 14.

The course of the test was recorded by a pair of cameras and the evaluation of deformation was performed based on the so-called photogrammetric method—generally speaking, “finding information from photographs.” In order to acquire and record the surface of the specimen during the hydraulic bulge test, it was necessary to apply a so-called pattern on its surface. This was carried out by using a combination of a white background and a black random spraying. This pattern allows the cameras to identify and acquire areas of a given size (referred to as facets), based on a certain assigned greyscale (ratio of white to black). Sizes of these facets are expressed in pixels. These facets are subsequently scanned and identified throughout the whole deformation process and, based on the change in their position, the deformation and kinematic quantities of the deformation process are calculated and evaluated. The basic evaluation of the acquired images and subsequent processing of data and characteristics were performed by the software Mercury RT v2.9 from the company Sobriety, see Figure 15. Subsequent work with the data, evaluation of measured dependencies and plotting the graphs were carried out in Origin v2020 software.

As a major output from this test, there is a dependence of effective (true) stress σ_EF_ vs. effective (true) strain *ϕ*_EF_ under the equi-biaxial loading. These values are obtained by substituting the measured parameters and quantities into Equations (21)–(23).
(21)$$\sigma_{EF}=\frac{pR}{2t}\;$$
(22)$$\;\;\varphi_{EF}=\frac{2\sqrt{3}}{3}\sqrt{\varphi_{1}^{2}+\varphi_{1}\varphi_{2}+\varphi_{2}^{2}}=\varphi_{3}\;$$
(23)$$t=t_{0}e^{\varphi_{3}}\;\;$$
where:*σ*_EF_—effective stress [MPa],*p*—hydraulic pressure [MPa], *φ*_EF_—effective strain [-],*R*—radius of curvature [mm],*φ*_1,2,3_—principal strains [-],*t*, *t*_0_—actual and initial thickness [mm].

### 3.3. Plane Strain Test

The plane strain test represents a special test, where the main condition is to ensure that deformation in the width direction equals zero. This is carried out by a specimen with a special shape, which is schematically illustrated in Figure 16. This test was again carried out on a TIRA Test 2300 testing device in a similar way to the static tensile test. Also, in this case, the specimen is loaded by the uniaxial tensile load and the whole test is carried out up to material failure (crack occurrence in the notch area).

The basic deformation condition (arising from the constant volume law), which must be met to carry out the plain strain test correctly and determine the valid results, can be characterised by Equation (24). In addition to that, Equation (25) further defines the computation of the relevant principal strain-length direction in this case.
(24)$${{\varphi_{1}=\varphi }_{3},\;\varphi }_{2}=0$$
(25)$$\;\varphi_{1}=ln\frac{L}{L_{0}}\;$$
where:*φ*_1_—principal strain (length) [-],*φ*_2_—principal strain (width) [-],*φ*_3_—principal strain (thickness) [-],*L*—actual length [mm],*L*_0_—initial length [mm].

The whole course of the test on a TIRA Test 2300 testing device was recorded using a load cell (force sensing) and a contact-less optical system from the company Sobriety that analysed deformation in the loaded area. The realisation of the plane strain test is shown below in Figure 17.

The data recording and basic evaluation of the measured quantities were again performed in the software Mercury RT (see Figure 18), from which we obtained the basic deformation characteristics of the material and the force records. These deformation characteristics were again determined via the photogrammetric method. Subsequent processing of data, evaluation of the measured dependencies and plotting the graphs were again carried out in the software Origin 2020.

### 3.4. Shear Test (Slotted Shear Test)

The shear test is used to introduce the shear stress load in the formed material. During this test, the material in the form of a testing specimen is subjected to a shear load, which occurs here in the form of simple shear plane stress in one shear plane (American standard ASTM B831 [48]). This stress state is in this case achieved by means of different geometries of the testing specimen, which are schematically illustrated in Figure 19. This testing specimen is loaded with a continuously increasing force induced by the translational movement of the clamping jaws. The test is carried out until specimen failure, i.e., crack occurrence in the shear plane of the testing specimen.

The shear test was again carried out on the TIRA Test 2300 testing device, where the test specimen was clamped by the mechanical wedge action grips and subjected to loading. Also, in this case, the whole course of the test was recorded both by a load cell mounted on the crossbar of the testing device and the contact-less optical system Mercury RT. An example of the shear test realisation is shown in Figure 20.

The recording and basic evaluation of the determined quantities were again carried out in the software Mercury RT (see Figure 21), from which we obtained the basic deformation characteristics of the material, as well as the force records. The aforementioned deformation characteristics were again obtained based upon the photogrammetric method (already described above). Subsequent work with the data, evaluation of the measured dependencies and plotting the graphs were again carried out in the software Origin 2020.

### 3.5. Cyclic Test (Fully Reversed Alternating Cycle)—Stress Ratio R = −1

The last of the needed material tests was a fully reversed alternating cycle test (stress ratio R = −1). This test is used to define the kinematic hardening of the tested material during its deformation. In this test, fully alternating tensile and compressive loading is applied (see Figure 22). As a result of this loading, the so-called Bauschinger effect takes place in the deformed material, i.e., there is change in the yield strength value due to the used alternating loading.

This test was again carried out on the TIRA Test 2300 testing device. Here, the geometry of the testing specimen corresponds to the static tensile test. The specimen was clamped by the special hydraulic jaws which, due to their shape (corresponding to the specimen geometry), allowed stabilisation of the specimen against buckling and its collapse during the compressive loading (see Figure 23). The course of the test was recorded by a load cell that measured the loading force and an axial extensometer, which determined the absolute elongation of the testing specimen during tensile and compressive loading.

The evaluation of basic quantities and parameters was again performed using the software LabNET associated with the TIRA Test 2300 testing device. Subsequent post-processing of data, evaluation of the measured dependencies and graphing were again carried out in Origin 2020 software. The cyclic loading test resulted in a dependence of the true stress vs. true strain, which created a hysteresis loop diagram.

### 3.6. Preparation of the Real Stamping Corresponding to the Process Set in Numerical Simulation

To verify and prove the validity of results obtained from the numerical simulation, it was necessary to produce a real stamping that could be compared with the results of the numerical simulation. Here, the process of drawing the sheet strip over the draw-bead was chosen. The test was performed on the tribological device Sokol 400, which is commonly used for tribological tests. It uses hydraulic grips and a moveable crossbeam allows drawing of a sheet strip between the tribological jaws, which can also include the draw-bead. A schematic illustration of this drawing process is shown in Figure 24.

This experiment was carried out on the testing jaw with a draw-bead, where the tensile and compressive stresses on the material change during drawing of the sheet strip. Thus, the Bauschinger effect is applied here, as in the real process of stamping shaped parts. The producing of the real stampings was in this case carried out for a feed of 100 mm and a feed rate of 50 mm/min. This real stamping prepared by the strip drawing over the draw-bead is shown in Figure 25.

The final specimens were finally removed from the tribological tool and subsequent material spring-back had to be subjected to shape analysis to determine the resulting contour of the given part. This was carried out using a 3D SOMET XYZ 464 coordinate measuring machine and the software TANGO!3D (see Figure 26). The obtained contour of the real stamping in the form of measured points (curves) was subsequently matched with the relevant area of the given sheet strip width in the software CATIA V5- 6R2019 to create a digital form of the real stamping. This digital form was further compared with the results of the numerical simulation in the software PAM-STAMP 2G 2015.

## 4. Results

### 4.1. Mechanical Testing of TRIP Steel HCT690

In Figure 27 are shown the final courses of stress–strain curves from the static tensile test for the individual rolling directions—Figure 27a. Moreover, in Figure 27b is shown the Krupkowski approximation of the stress–strain curve for the reference direction 0°, which was used to determine the approximation coefficients *C*, *n* and *φ*_0_.

In Table 1 are summarised the basic mechanical properties of the tested material HCT690 measured by the static tensile test. Table 2 shows the approximation constants determined by the Krupkowski approximation and also normal anisotropy coefficients from the static tensile test.

In Figure 28 are shown the final courses of stress–strain curves from the equi- biaxial loading of the material—Figure 28a. Moreover, in Figure 28b is shown the Krupkowski approximation of the stress–strain curve that was used to determine approximation coefficients *C*, *n* and *φ*_0_. These are summarised in Table 3, together with the normal anisotropy coefficient from the equi-biaxial test.

In Figure 29 are shown the final courses of stress–strain curves from the plane strain test for the individual rolling directions—Figure 29a. Moreover, in Figure 29b is shown the Krupkowski approximation of the stress–strain curve for the reference direction 0°, which was used to determine the approximation coefficients *C*, *n* and *φ*_0_. These are summarised in Table 4 with respect to rolling direction.

Figure 30 shows the final courses of stress–strain curves from the shear test for the individual rolling directions—Figure 30a. Moreover, in Figure 30b is shown the Krupkowski approximation of the stress–strain curve for the reference direction 0°, which was used to determine the approximation coefficients *C*, *n* and *φ*_0_. These are summarised in Table 5 with respect to rolling direction.

In Figure 31 is shown the final course of the stress–strain curve for cyclic loading in a fully reversed alternating cycle.

### 4.2. Definition of the Used Yield Criteria in the Numerical Simulation Environment of the Software PAM-STAMP 2G

Definition of the material models in PAM-STAMP 2G software is based on the so- called data “fitting.” The data and needed material characteristics have to be implemented in a certain way in the numerical simulation. On the material card, the basic material properties must be entered and the material computational model must be selected as well. Moreover, relevant variables for defining the relevant yield criteria must be entered as well. Finally, there is a need to select and define the material hardening law. The data for definition of the material model in PAM-STAMP 2G are included in Table 6, Table 7 and Table 8.

Figure 32 shows the definition of used yield criteria: Hill 48 (left), Vegter Lite (middle) and Vegter Standard (right). Here, the basic material properties are entered, the relevant yield criterion are selected and the mechanical material values obtained via the individual material tests are entered as well. The stress characteristics are in this case always entered as the ratio of the given stress from the selected test and selected rolling direction to the reference stress value obtained from the static tensile test in the reference rolling direction 0°.

In Figure 33 is illustrated the definition of the isotropic hardening model using the approximation constants *C*, *n* and *φ*_0_ obtained from the Krupkowski approximation of the static tensile test in the reference direction 0°. In Figure 34 is subsequently shown so-called data “fitting” in the software MatPara v2.1.0.0 to determine parameters for the proper definition of the Yoshida kinematic hardening model.

### 4.3. Numerical Simulation of the Sheet Metal Forming Process

Using numerical simulation in the software PAM-STAMP 2G, the sheet metal forming process corresponding to the real experiment was carried out. Specifically, a tribological strip drawing test was carried out, also using so-called draw-beads. The calculation of the numerical simulation was carried out concerning the selected material model and the material hardening law during deformation. The deformation process was simulated as follows—the tool closing first, followed by the drawing of the metal strip over the draw-bead (Figure 35) and finally the material spring-back was simulated (Figure 36).

## 5. Discussion

In the following subsections, the effects of selected yield criteria on the calculation of the forming process and subsequent material spring-back are evaluated. The individual results of numerical simulations, as well as results from the real experiments taking into account the stamping, are described and compared here, always with respect to the chosen yield criterion, including the choice of the material hardening law during deformation.

### 5.1. Comparison of the Yield Criteria Used in the Numerical Simulation

Concerning the performed material tests, the results of the individual selected yield criteria used in the numerical simulation for calculation of the deformation process and subsequent spring-back of the material were compared. The following yield criteria were used: Hill 48, Vegter Lite and Vegter (sometimes marked as Vegter Standard). These yield criteria define the so-called plasticity boundaries employing the individual material parameters, which determine the position of the relevant reference (control) points of the ellipse representing the relevant yield criterion on the planes *σ*_1_ and *σ*_2_.

Figure 37 shows a comparison of selected yield criteria for material HCT690. When comparing these computational models, it is possible to observe the variation in the position of relevant plasticity boundaries between these criteria. The model Hill 48 is shown here only as an indicator for comparison. This model requires the least demanding conditions because in this case only data from the static tensile test are needed. In the case of the more complex yield criteria—Vegter Light and Vegter Standard—the position of the plasticity boundary is further influenced by additional material tests under the defined loading method (described above). The plasticity boundary represents the transition from an elastic to plastic state and its position directly influences the portion of elastic deformation in the formed material. That is why it is so important for subsequent calculation of the spring-back.

### 5.2. Comparison of the Results from the Numerical Simulation and the Real Stamping

Figure 38 shows a final comparison of selected yield criteria under the variant with isotropic hardening of the material during deformation. The yield criterion Hill 48 with isotropic hardening (magenta), Vegter Lite with isotropic hardening (green) and Vegter Standard with isotropic hardening (blue) are compared. It is possible to observe quite significant deviations between the contours obtained by numerical simulation for each yield criterion and the contour of the sheet strip obtained from the real process of strip drawing over the draw-bead. When comparing the individual contours, it can be observed that the isotropic hardening law fails to adequately follow and describe the magnitude of the spring-back in this deformation process. In Figure 39, differences between the real sheet contour and contour from the numerical simulation acc. to Hill 48 isotropic hardening law are shown, which in this case is the closest one to the real stamping.

Subsequently, in Figure 40, a comparison of selected yield criteria under the variant with kinematic hardening law is shown. This shows again a comparison of the sheet contours obtained via calculation for the Hill 48 (magenta), Vegter Lite (green) and Vegter Standard (blue) models with kinematic hardening during the deformation of material. From this comparison, it is possible to observe the benefit of the kinematic hardening law during deformation, which is already able to better follow and describe the shape and magnitude of the spring-back. This is due to the fact that, as material passes through the draw-bead, in the deformed material a repeated change of the state of stress and strain takes place. This results in the dominant manifestation of the Bauschinger effect, which the kinematic hardening law can take into account—in contrast to the isotropic hardening law. The best results here are achieved when the kinematic hardening law is used in combination with the Vegter Standard yield criterion. In Figure 41 are shown histograms, which quantify the differences between the real sheet contour and the most accurate contour from the numerical simulation acc. to Vegter Standard kinematic hardening law.

The numerical computation using the finite element method is based on mathematical assumptions, which always introduce a certain degree of distortion in the numerical computation compared with reality. This mainly involves replacing the continuous environment of the formed material by a finite element mesh of a given size, which affects the accuracy of the numerical simulation. Another possible error in the computation is given by the chosen numerical model, which takes into account the assumption of a constant friction coefficient, as opposed to the real experiment, where the actual magnitude of this friction coefficient may vary due to the changes in the contact pressure, e.g., on the draw-beads. Another possible distortion of the numerical computations compared to reality is the consideration that the computation does not result in a deviation of the normal to the finite element due to shear stresses (the so-called Mindlin hypothesis). This assumption may not be met in a real experiment, when the sheet is repeatedly bent over the small radii of the draw-beads. Another possible deviation is certainly the change in the tensile modulus E, when the loading direction (tension vs. compression) changes in the opposite direction during bending on the draw-bead.

The major contribution of this paper and the determined results is certainly the fact that these results can generally improve the refinement of the numerical simulation of sheet metal forming and the subsequent spring-back of material. These results can further support a possible improve in production efficiency and can reduce the cost of the real stamping process and decrease individual production times. In practice, these results and conclusions are of further importance in terms of the choice of possible solutions for selected types of material or different geometries of the manufactured parts. From the scientific contribution point of view, these results bring further useful aspects and contributions to the research and understanding of the given issue in light of materials engineering, material testing and numerical simulations of the sheet metal forming process. Furthermore, it is expected that these results will serve as a basis and data support for further research and development in the solution of the given problem in the field of material properties analysis as well as the mathematical modelling of the forming process in numerical simulations.

## 6. Conclusions

This paper was focused on the research and analysis of mechanical properties and stress–strain behaviour of the tested material—TRIP steel HCT690. Our research dealt with the possibility of using and applying these determined material characteristics in the numerical simulations of the sheet metal forming process. Mechanical properties and material characteristics were determined using selected material tests, i.e., tests considering the required states of stress needed to define the relevant yield criterion boundary and subsequent deformation behaviour in the region of severe plastic deformation. The measured material parameters and dependencies were further used to define material models in a numerical simulation environment and to simulate the subsequent process of the strip drawing over the draw-bead as well as the subsequent spring-back of the material in the software PAM-STAMP 2G. To evaluate and compare the results of the numerical simulation, a real stamping was also prepared using the same process parameters as in the numerical simulation. 

In terms of the qualitative evaluation of the research, the following conclusions can be drawn. It is not necessary to use advanced computational models to predict the formability (manufacturability) of a given part, which significantly increase the computational time but do not significantly increase the accuracy of the deformation analysis in terms of the magnitude of deformations and their distribution on the stamping. The position of deformation coordinates shown in the FLD is virtually identical when using both standard and advanced computational models. However, stamping in the zone of safe (allowable) deformations is not the only condition for production of stamping. Another factor influencing production is achieving the desired shape of the stamping within the tolerance limits, which is directly affected by the stamping shape stability after the forming process. In addition to the technological parameters influencing the forming process, the final shape of the given stamping is also greatly influenced by its spring-back. The presented research has shown that the choice of the computational model has a major influence on the prediction of the stamping spring-back and thus on the prediction of its shape changes after the forming process. 

A sufficiently accurate prediction of changes in the stamping shape is a prerequisite for the correct design and production of the pressing tools’ functional surfaces. This is the so-called shape compensation of these functional surfaces of tools compared to the theoretical CAD surfaces of the relevant stamping. It turns out that with the increasing shape complexity of stamping and utilisation of materials with a higher spring-back proportion (e.g., Al, Ti alloys, high-strength materials), the proper choice and use of the advanced computational models for the metal forming numerical simulations are crucial.

From the performed analyses, experimental tests and numerical simulations, it is clear that the proper selection of the relevant yield criterion (material calculation model) in the numerical simulation greatly influences the final accuracy of calculated deformation and especially the prediction of material spring-back in the numerical simulation. In the case of comparison of the individual yield criteria Hill 48, Vegter Lite and Vegter Standard, differences in the position of the yield criterion boundary can be observed, which to some extent directly affect the calculation of deformation and subsequent spring-back. Furthermore, it was found that the choice of material hardening law during deformation has a major influence and importance in the proper prediction of the relevant magnitude of spring-back. It has been found and proven that to obtain the most accurate results of the deformation behaviour and subsequent material spring-back, it is necessary to choose a kinematic hardening law for material deformation. From these results, it can be concluded that the best agreement of the numerical simulation with respect to the real process was achieved when using the kinematic “Yoshida” hardening law in combination with the most complex yield criterion—Vegter Standard.

## Figures and Tables

**Figure 1 materials-17-00535-f001:**
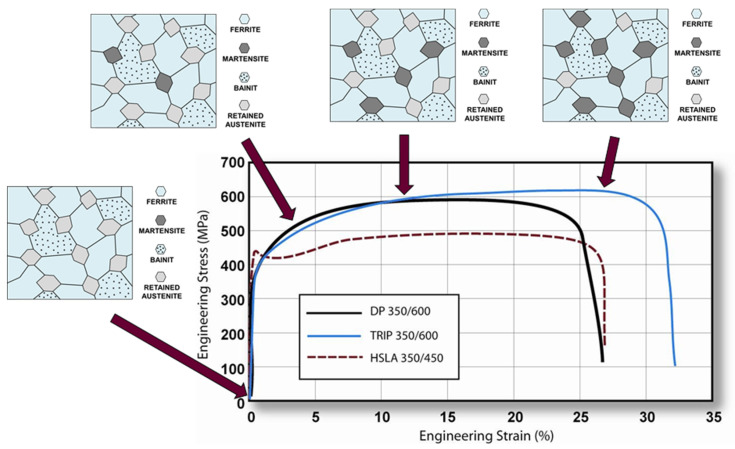
Illustration of the structure change of TRIP steel during deformation [36].

**Figure 2 materials-17-00535-f002:**
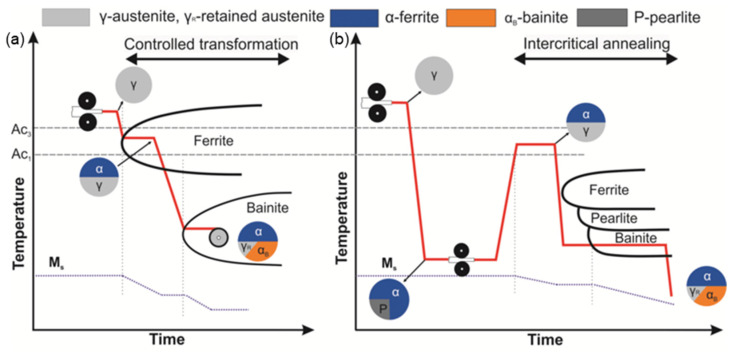
Schematic diagram of TRIP steel production process after hot rolling (**a**) and after cold rolling (**b**) [39].

**Figure 3 materials-17-00535-f003:**
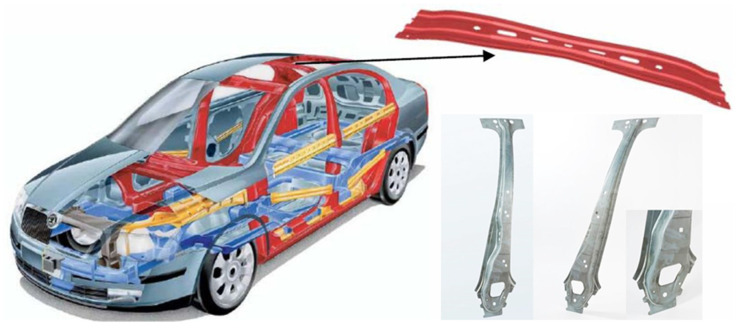
Examples of using TRIP steel on car body [36].

**Figure 4 materials-17-00535-f004:**
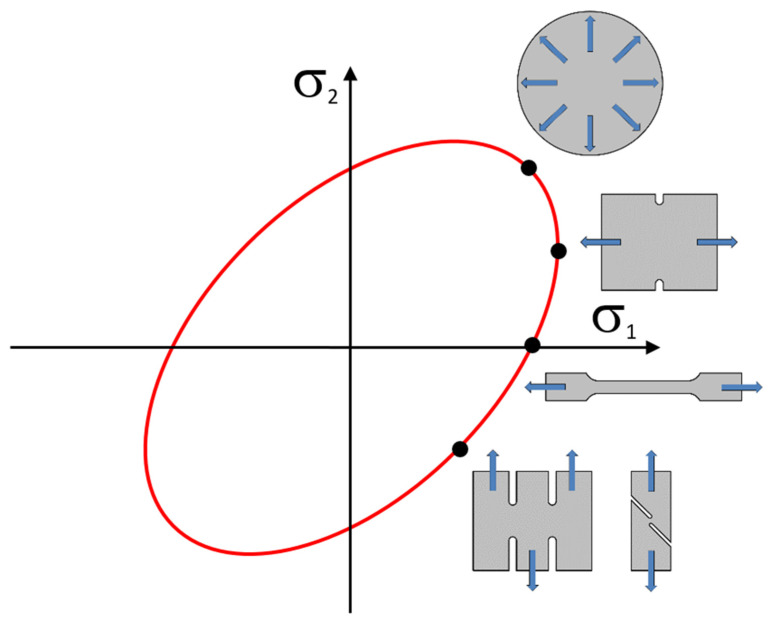
Reference (control) points of ellipse given by the relevant material tests.

**Figure 6 materials-17-00535-f006:**
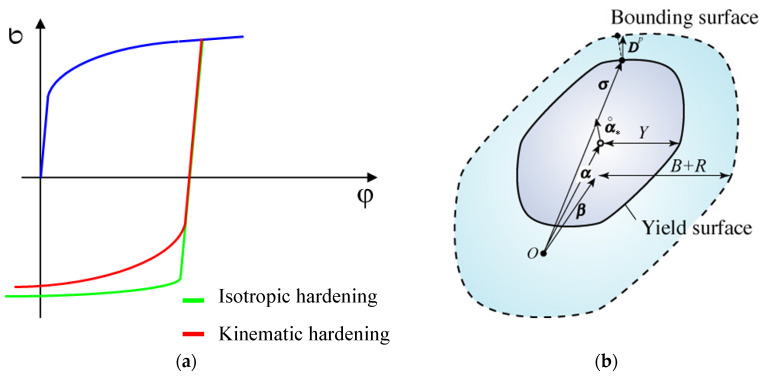
Illustration of the kinematic hardening (**a**) and shift of Yoshida yield surface (**b**) [46].

**Figure 7 materials-17-00535-f007:**
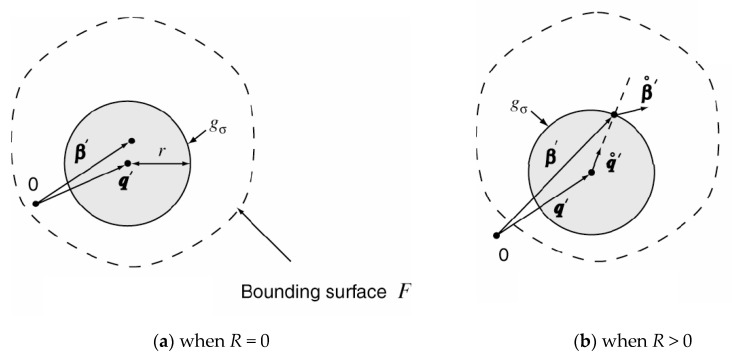
Definition of the non-isotropic hardening (non-IH) in the deviatoric stress space [46].

**Figure 10 materials-17-00535-f010:**
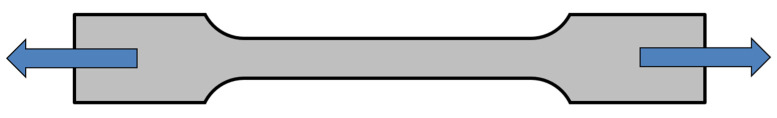
Schematic diagram of the specimen loading during the static tensile test.

**Figure 11 materials-17-00535-f011:**
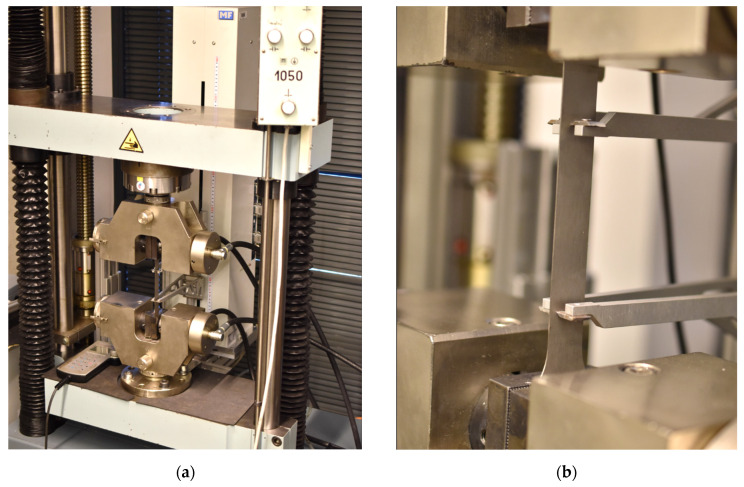
Realisation of the static tensile test by TIRA Test 2300 testing device (**a**) and the detail of the measured specimen (**b**).

**Figure 12 materials-17-00535-f012:**
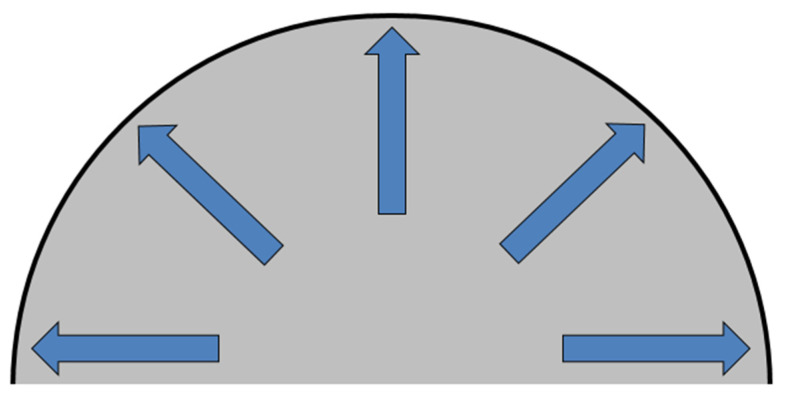
Schematic diagram of the specimen loading during the hydraulic bulge test.

**Figure 13 materials-17-00535-f013:**
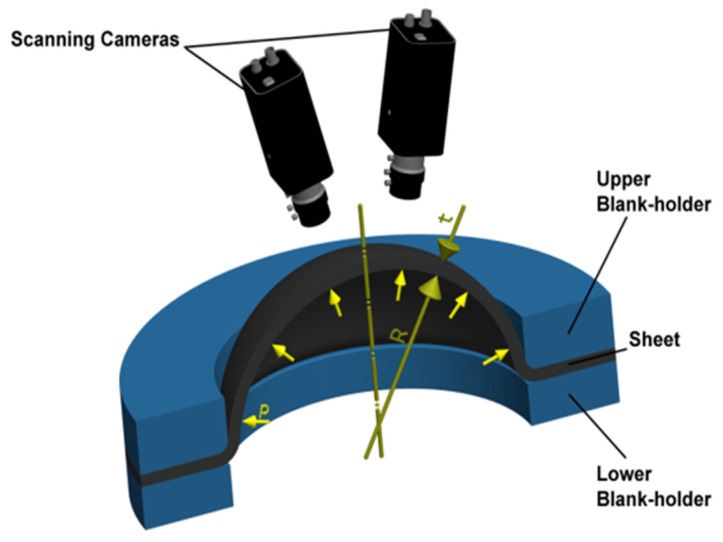
Principle of the hydraulic bulge test.

**Figure 14 materials-17-00535-f014:**
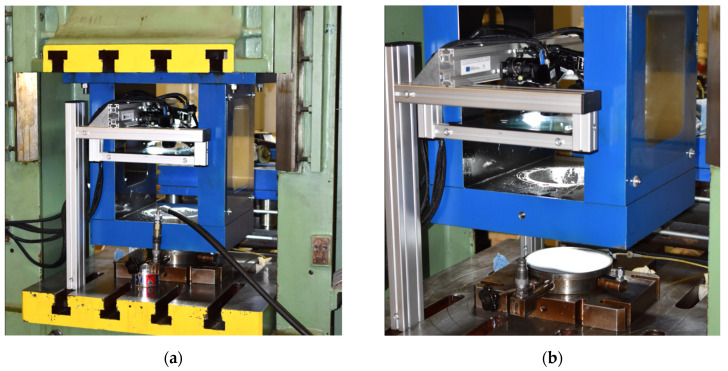
Realisation of the hydraulic bulge test on a CBA 300/63 hydraulic press (**a**) and the detail of the measuring device (**b**).

**Figure 15 materials-17-00535-f015:**
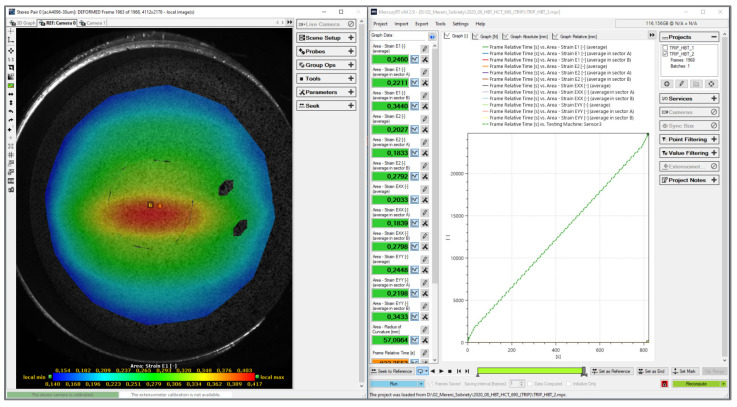
Environment of the software Mercury RT v2.9 during evaluation hydraulic bulge test.

**Figure 16 materials-17-00535-f016:**
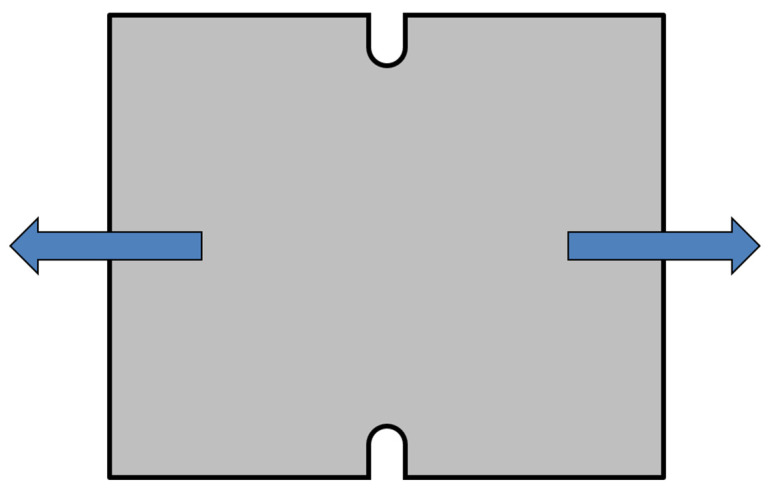
Schematic diagram of the specimen loading during the plain strain test.

**Figure 17 materials-17-00535-f017:**
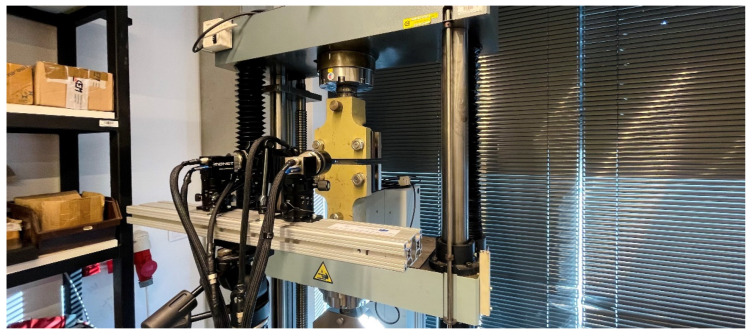
Realisation of the plane strain test on a TIRA Test 2300 testing device.

**Figure 18 materials-17-00535-f018:**
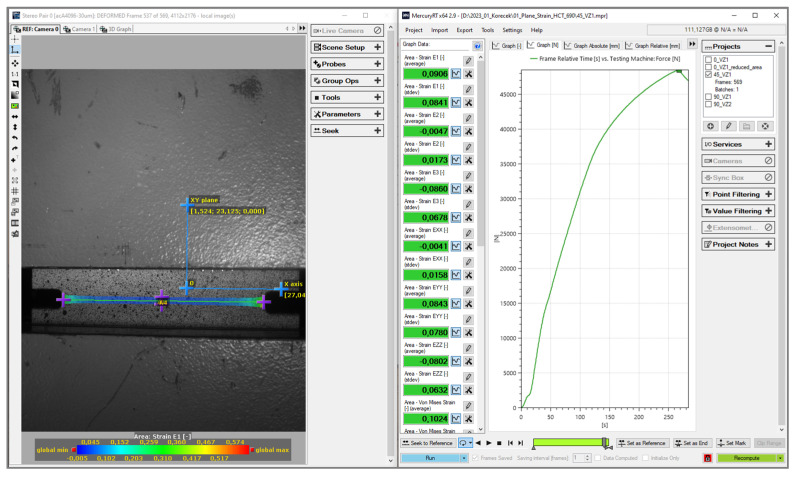
Environment of the software Mercury RT during evaluation of the plane strain test.

**Figure 19 materials-17-00535-f019:**
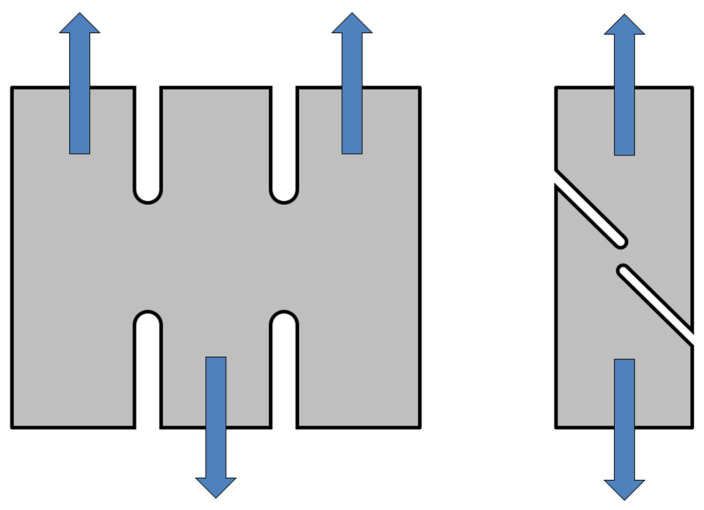
Schematic diagram of the specimen loading during the shear test.

**Figure 20 materials-17-00535-f020:**
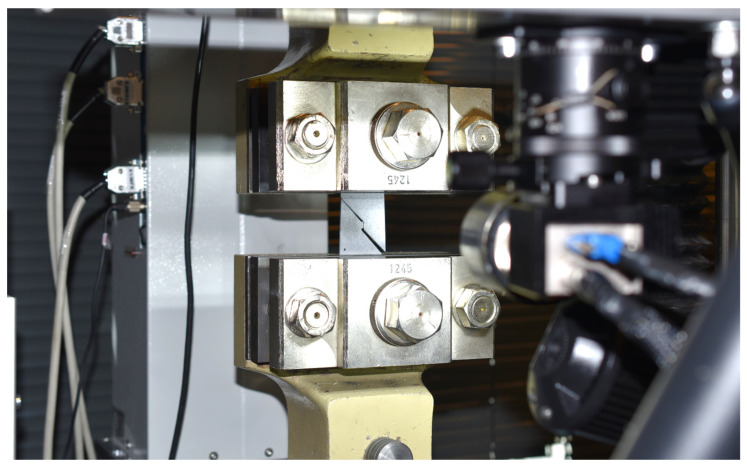
Realisation of the shear test on a TIRA Test 2300 testing device.

**Figure 21 materials-17-00535-f021:**
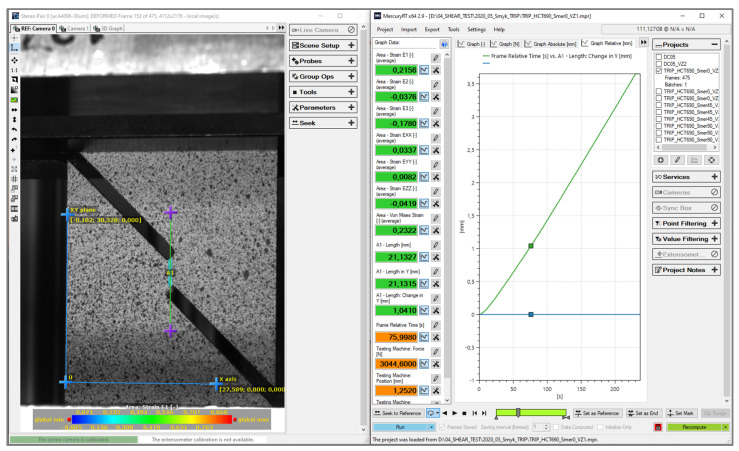
Environment of the software Mercury RT v2.9 during evaluation of the shear test.

**Figure 22 materials-17-00535-f022:**
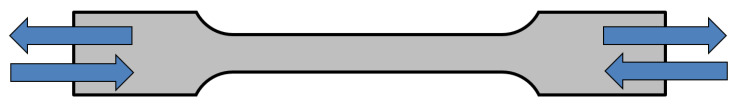
Schematic diagram of the specimen loading during the cyclic test.

**Figure 23 materials-17-00535-f023:**
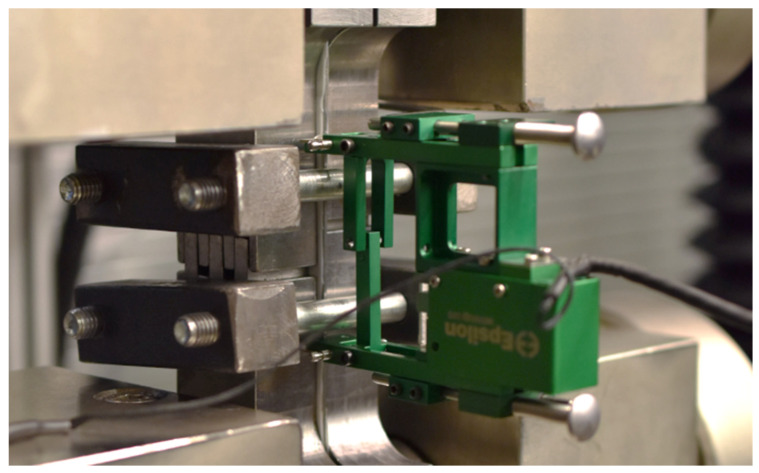
Realisation of the cyclic test on a TIRA Test 2300 testing device.

**Figure 24 materials-17-00535-f024:**
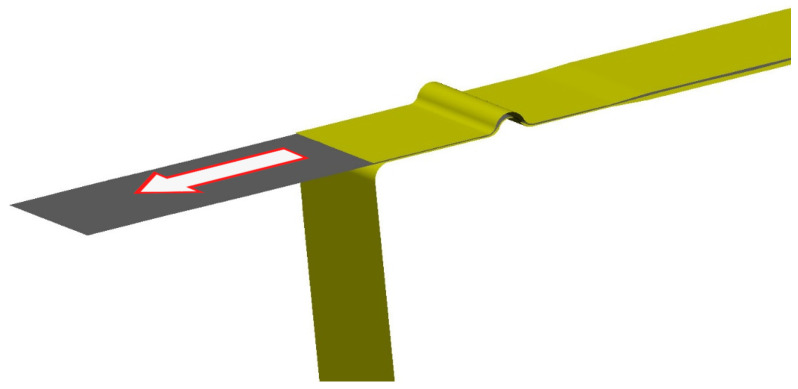
Schematic illustration of the sheet strip drawing over the draw-bead.

**Figure 25 materials-17-00535-f025:**
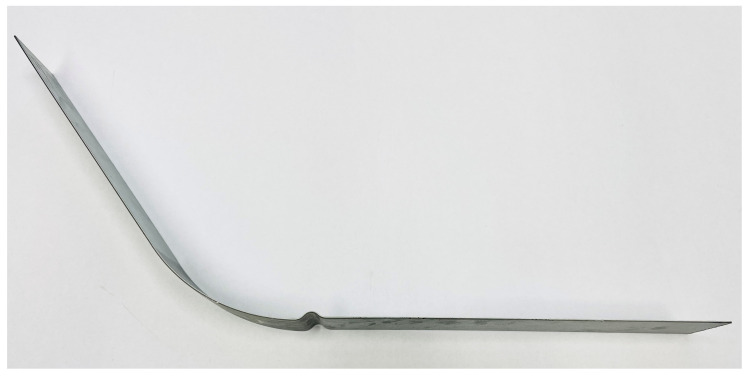
Specimen (real stamping) prepared by the strip drawing over the draw-bead.

**Figure 26 materials-17-00535-f026:**
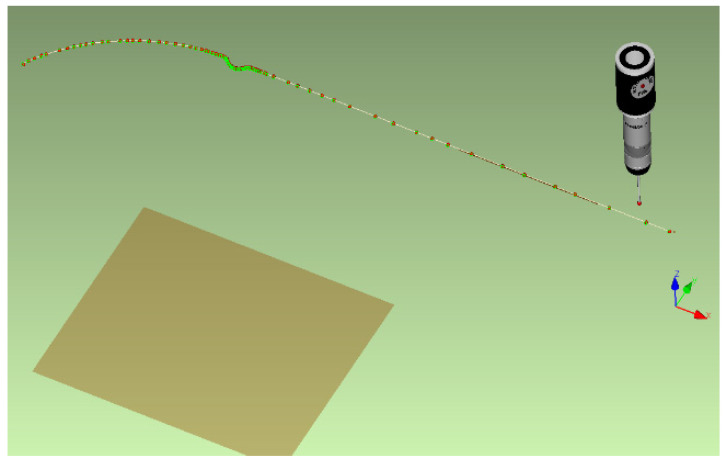
Measurement of the real stamping contour on a 3D coordinate measuring machine.

**Figure 27 materials-17-00535-f027:**
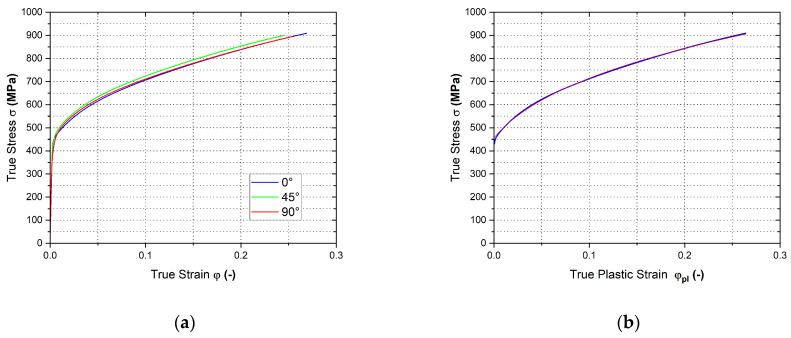
Stress–strain curve of the material HCT690 from the static tensile test (**a**) and Krupkowski approximation of the stress–strain curve in the rolling direction 0° (**b**).

**Figure 28 materials-17-00535-f028:**
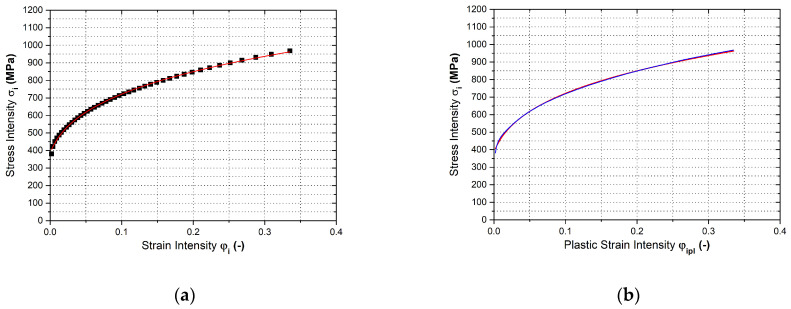
Stress–strain curve of the material HCT690 from the equi-biaxial test (**a**) and Krupkowski approximation of the stress strain curve (**b**).

**Figure 29 materials-17-00535-f029:**
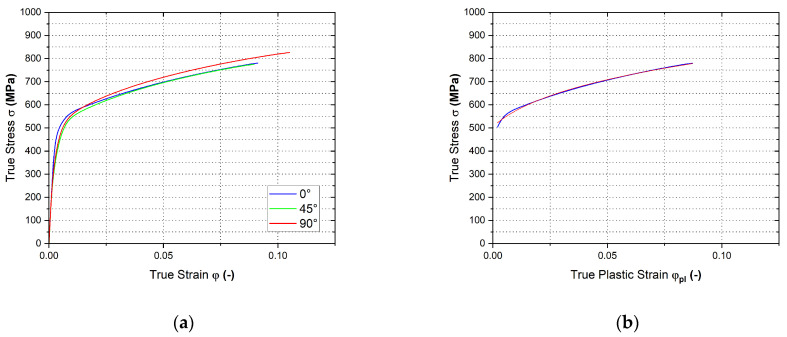
Stress–strain curve of the material HCT690 from the plane strain test (**a**) and Krupkowski approximation of the stress–strain curve in the rolling direction 0° (**b**).

**Figure 30 materials-17-00535-f030:**
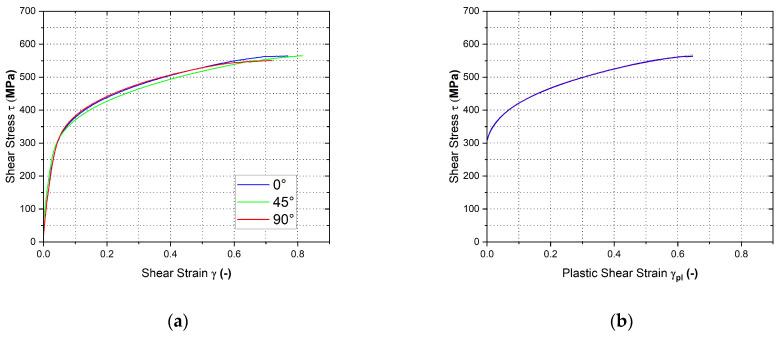
Stress–strain curve of the material HCT690 from the shear test (**a**) and Krupkowski approximation of stress–strain curve in the rolling direction 0° (**b**).

**Figure 31 materials-17-00535-f031:**
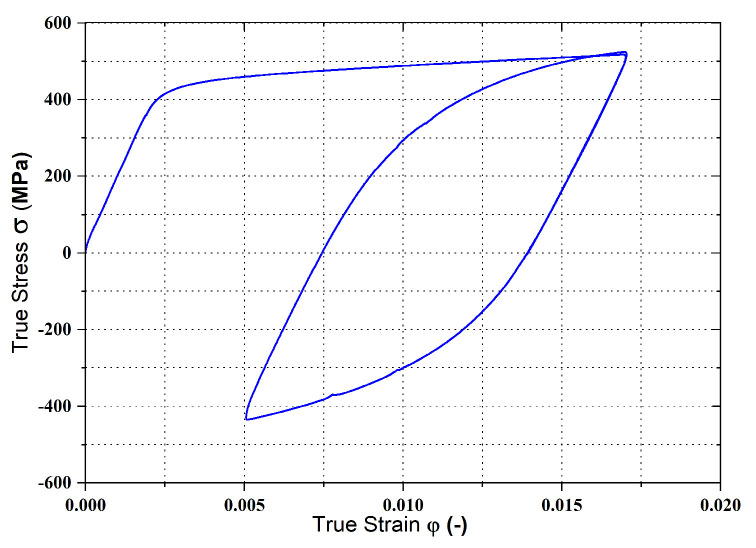
Stress–strain curve of the material HCT690 from the cyclic test in the rolling direction 0°.

**Figure 32 materials-17-00535-f032:**
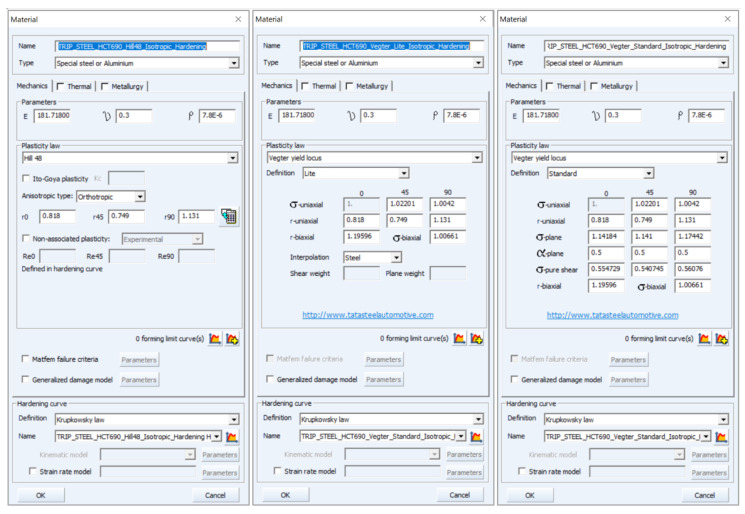
Definition of the yield criterion: Hill 48 (**left**), Vegter Lite (**middle**) and Vegter Standard (**right**) for material HCT690.

**Figure 33 materials-17-00535-f033:**
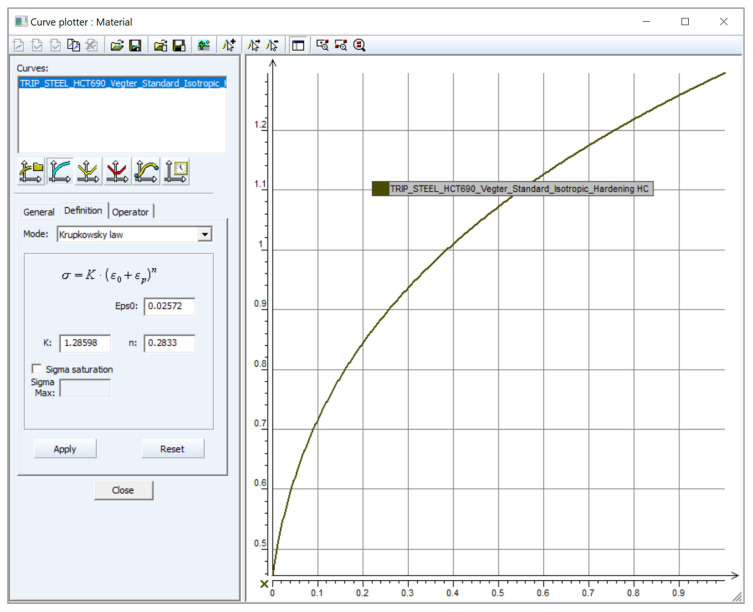
Definition of the isotropic hardening law in the PAM-STAMP 2G software for material HCT690.

**Figure 34 materials-17-00535-f034:**
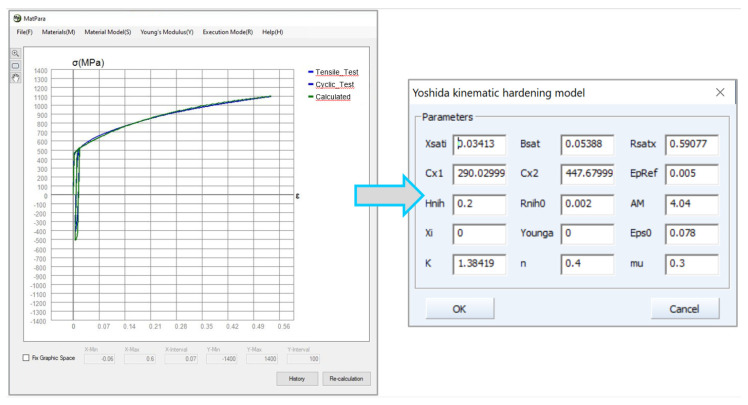
Filtering of parameters for Yoshida model in MatPara software for material HCT690.

**Figure 35 materials-17-00535-f035:**
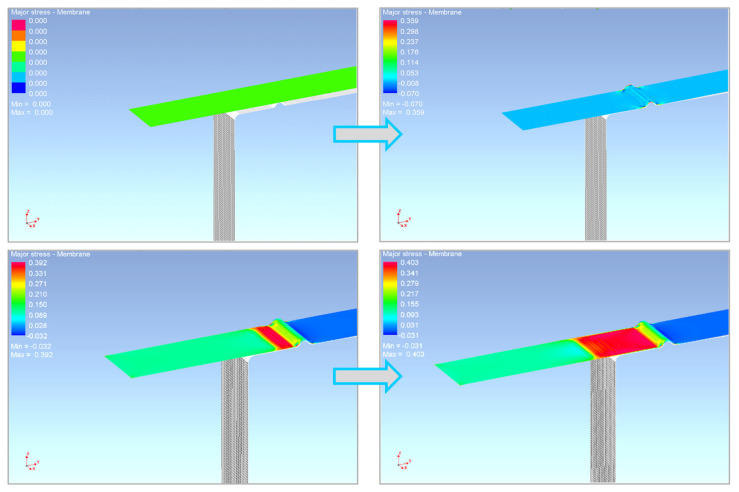
A course of the numerical simulation process in the environment PAM-STAMP 2G.

**Figure 36 materials-17-00535-f036:**
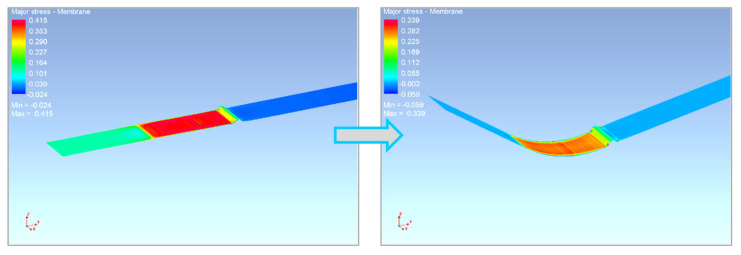
Spring-back of the given part in the numerical simulation in PAM-STAMP 2G.

**Figure 37 materials-17-00535-f037:**
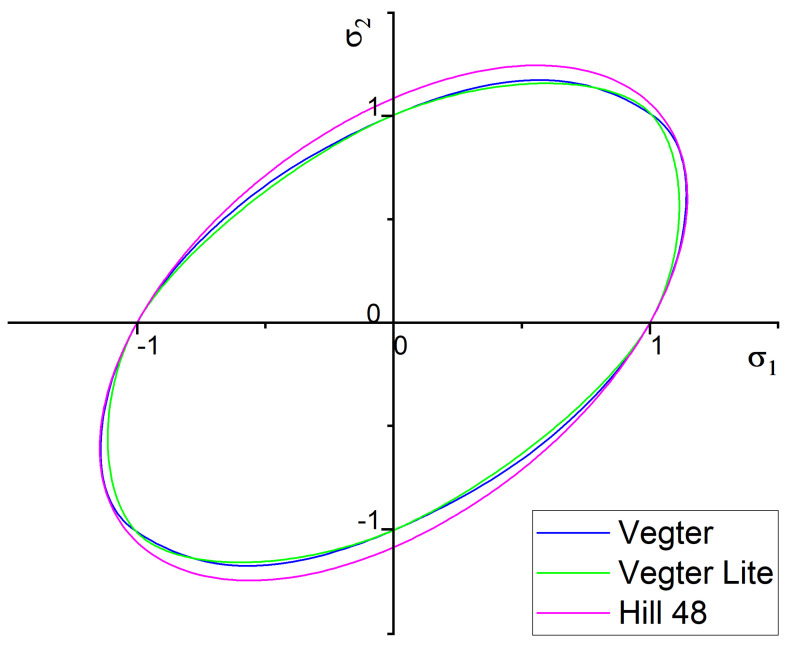
Comparison of the selected material models representing the yield criteria in the planes *σ*_1_ and *σ*_2_ for material HCT690.

**Figure 38 materials-17-00535-f038:**
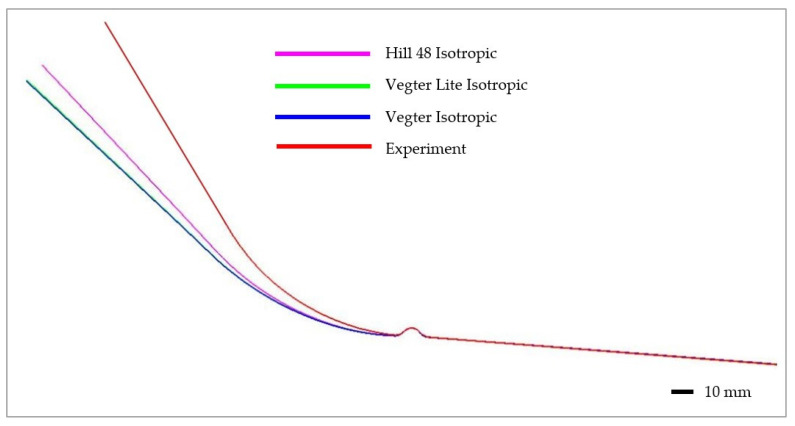
Comparison of the resulting contour obtained by numerical simulation of Hill 48, Vegter Lite and Vegter Standard with isotropic hardening law for material HCT690.

**Figure 39 materials-17-00535-f039:**
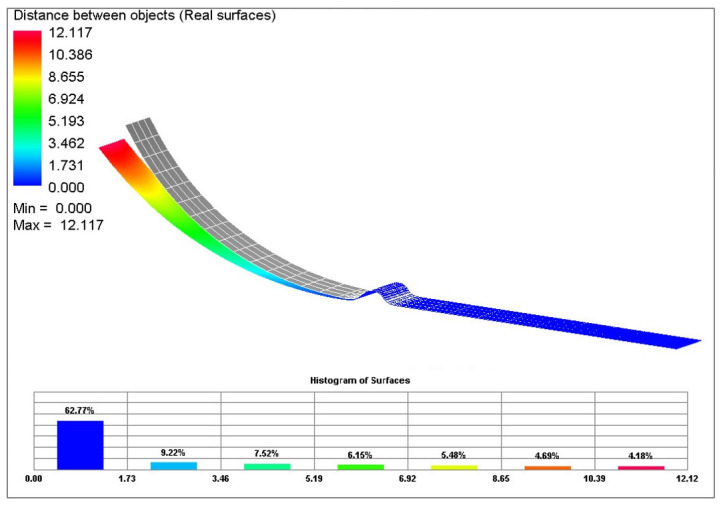
Difference between the sheet contour from Hill 48 isotropic hardening law and the real contour of the given stamping.

**Figure 40 materials-17-00535-f040:**
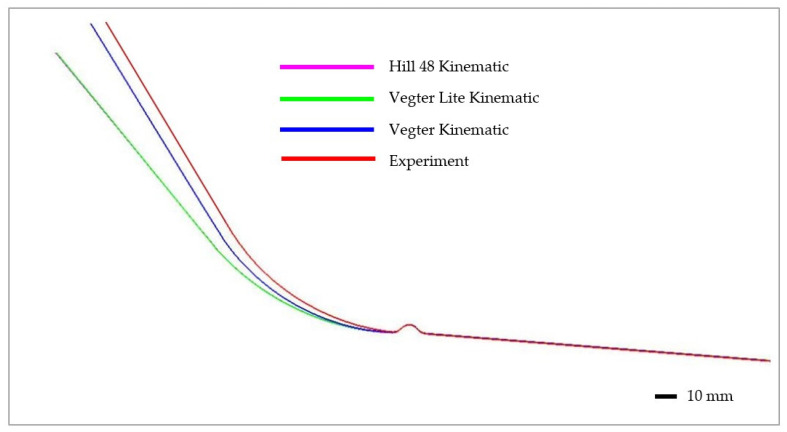
Comparison of the resulting contour obtained by numerical simulation of Hill 48, Vegter Lite and Vegter Standard with kinematic hardening law for material HCT690.

**Figure 41 materials-17-00535-f041:**
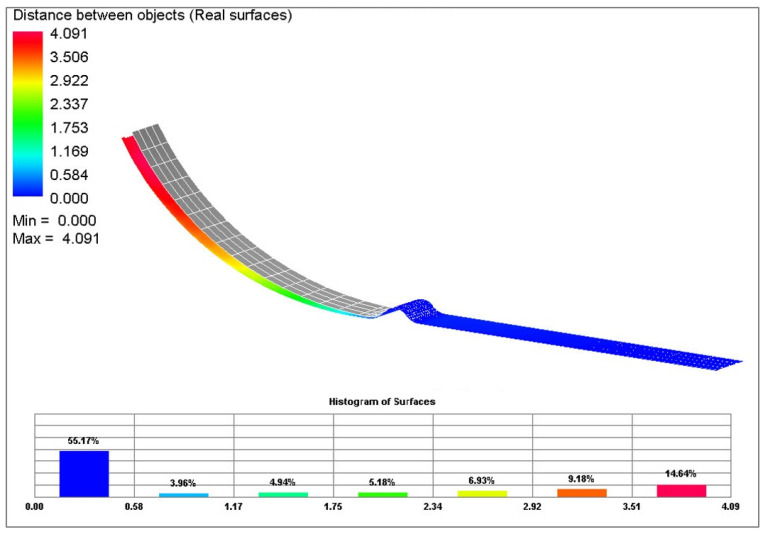
Difference between the sheet contour from Vegter Standard kinematic hardening model and the real contour of the given stamping.

**Table 1 materials-17-00535-t001:** Basic mechanical properties of material HCT690.

Rolling Direction (°)	*R*_p0,2_ (MPa)	*R*_m_ (MPa)	*A*_g_ (-)	*A*_80mm_ (-)	*E* (MPa)
0	456.90 ± 1.05	695.09 ± 1.10	0.3086 ± 0.0022	0.3745 ± 0.0038	181.718 ± 112
45	457.65 ± 0.94	704.43 ± 1.22	0.2787 ± 0.0028	0.3258 ± 0.0034	194.229 ± 136
90	431.64 ± 1.12	694.32 ± 1.06	0.2896 ± 0.0018	0.3378 ± 0.0042	188.768 ± 123

**Table 2 materials-17-00535-t002:** Approximation constants determined by the Krupkowski approximation of the stress–strain curve from the static tensile test for material HCT690.

Rolling Direction (°)	C (MPa)	n (-)	ϕ_0_ (-)	*R* (-)
0	1285.9839 ± 0.08008	0.28330 ± 5.23372 × 10^−5^	0.02572 ± 1.79626 × 10^−5^	0.8180 ± 0.012
45	1262.0275 ± 0.11845	0.25529 ± 6.89821 × 10^−5^	0.01914 ± 2.14572 × 10^−5^	0.7490 ± 0.009
90	1235.1558 ± 0.15673	0.25001 ± 9.10215 × 10^−5^	0.01647 ± 2.71825 × 10^−5^	1.1310 ± 0.014

**Table 3 materials-17-00535-t003:** Approximation constants determined by the Krupkowski approximation of the stress–strain curve from the equi-biaxial test for material HCT690.

Rolling Direction (°)	C (MPa)	n (-)	φ_0_ (-)	*R* (-)
-	1257.7543 ± 6.08372	0.25056 ± 0.00307	0.00887 ± 6.79748 × 10^−4^	1.1960 ± 0.015

**Table 4 materials-17-00535-t004:** Approximation constants determined by the Krupkowski approximation of the stress-–strain curve from the plain strain test for material HCT690.

Rolling Direction (°)	C (MPa)	n (-)	φ_0_ (-)
0	1224.6926 ± 6.87837	0.19513 ± 0.00250	0.01066 ± 4.26765 × 10^−4^
45	1138.8545 ± 5.55614	0.16041 ± 0.00178	0.00219 ± 1.83297 × 10^−4^
90	1216.7021 ± 4.07405	0.17219 ± 0.00129	0.00274 ± 1.44064 × 10^−4^

**Table 5 materials-17-00535-t005:** Approximation constants determined by the Krupkowski approximation of the stress– strain curve from the shear test for material HCT690.

Rolling Direction (°)	C (MPa)	n (-)	φ_0_ (-)
0	609.8419 ± 0.45077	0.17484 ± 8.38446 × 10^−4^	0.02078 ± 5.00940 × 10^−4^
45	600.7655 ± 0.29863	0.18977 ± 7.81576 × 10^−4^	0.03279 ± 6.37659 × 10^−4^
90	598.0089 ± 0.53662	0.15590 ± 8.80404 × 10^−4^	0.01462 ± 4.52226 × 10^−4^

**Table 6 materials-17-00535-t006:** Basic physical properties of the TRIP steel HCT690.

Young Modulus E (GPa)	Poisson Ratio ν (-)	Density ρ (kg.m^−3^)
181.718	0.3	7.8 × 10^−6^

**Table 7 materials-17-00535-t007:** Anisotropy coefficients of the TRIP steel HCT690.

Direction 0 (-)	Direction 45 (-)	Direction 90 (-)	Biaxial (-)
0.818	0.749	1.131	1.19596

**Table 8 materials-17-00535-t008:** Stress ratio for the definition of the plasticity law.

Rolling Direction (°)	Uniaxial (-)	Plane (-)	Shear (-)	Biaxial (-)
0	1	1.14184	0.55473	
45	1.02201	1.14100	0.54075	1.00661
90	1.00420	1.17442	0.56076	

## Data Availability

Data are contained within the article.

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
