# Peer review of "Analysis of TRIP Steel HCT690 Deformation Behaviour for Prediction of the Deformation Process and Spring-Back of the Material via Numerical Simulation"

_materials, 2024, doi:10.3390/ma17030535_

Round 1
Reviewer 1 Report
Comments and Suggestions for Authors
(1) There are still some hard expressions.
(2) From Fig.38 and 39, how do you determine better description of the spring-back shape and magnitude? There are no any unit and scale. It is better to characterize the spring-back quantitively.
(3) It is essential to determine the inner mechanisms of the difference between experimental and simulation results.
Comments on the Quality of English LanguageThere are many grammar errors.
Author Response
Dear Reviewer 1,
you can fing our response in the attachment.

Reviewer 2 Report
Comments and Suggestions for Authors
The English language should be refined and corrected, and there are too many grammatical problems.
The article needs to be expanded with recent articles in the introduction section; only three articles are updated. The introduction should be lengthy sufficiently to illustrate the past work, and the writers should compare the other studies to the current study.
The conclusion section should be rewritten to include informative and qualitative information.
Figures 32, 33, 34 should be removed from the manuscript.
In the page 22 the authors write, “Results 534 The following chapters deal with the individual results of material testing with re- 535 spect to the performed tests, definition of the material model and the subsequent numer- 536 ical simulation of the selected forming process in the environment of software PAM- 537 STAMP 2G” I assume they copy paste their manuscript from the report. This writing method is extremely awful. This issue is repeated in every section throughout the work.
Figure 10 needs the dimension of the tensile test sample.
Material and methods should contain the experimental element, and I advocate developing a new section to cover the numerical simulation.
The current manuscript needs greater clarification about the outcomes from the experimental work and simulation and the comparison between them.
The manuscript is like a scientific report. The article needs many modifications to be in the form of scientific research.
Comments on the Quality of English Language
The English language should be refined and corrected, and there are too many grammatical problems.
Author Response
Dear Reviewer 2,
you can fing our response in the attachment.

Reviewer 3 Report
Comments and Suggestions for Authors
The article submitted for review entitled " Analysis of TRIP Steel HCT690 Deformation Behaviour for Prediction the Deformation Process and Spring-back of the Material by Numerical Simulation" raises the problem of determining the material characteristics in numerical simulations of the sheet metal forming process.
The findings obtained in the reviewed article on the basis of the experiment are a novelty:
1) The article considers the problem on the research and analysis of mechanical properties and stress-strain behaviour of the tested material – TRIP steel HCT690.
2) The content of the article is an original scientific approach and a kind of novelty in this issue.
3)The novelty that can be noticed in the publication is that the comparison the individual yield criterions Hill 48, Vegter Lite and Vegter Standard, differences in the position of the yield criterion boundary can be observed, which to some extent directly affects the calculation of deformation and subsequent spring-back. It was found that the choice of material hardening law during deformation has a major influence and importance on the proper prediction of the relevant magnitude of spring-back. The best agreement of the numerical simulation with respect to the real process was achieved when using the kinematic "Yoshida" hardening law in combination with the most complex yield criterion - Vegter Standard.
4) Research work pave the way for further development in the field of determined material characteristics in numerical simulations of the sheet metal forming process.
5) Probably (for the future) it would be appropriate to use computed tomography during the testing of recovered parts of fragments to analyze the structure in the comparison to simulations results. But at this stage of work, it is not necessary.
6) The conclusions are adequate with the evidences and research results presented in the article and they address to the main argument of the work. The authors of the article presented the results of their own research in a clear, transparent and exhaustive way.
7) All references are appropriate to the text.
8)The tables and figures are legible and clear:
Please correct:
· in line 527 – should be Figure 26;
· Table 4 and 5 are not mentioned in the text.
Author Response
Dear Reviewer 3,
you can fing our response in the attachment.

Reviewer 4 Report
Comments and Suggestions for Authors
The manuscript "Analysis of Deformation Behavior of TRIP HCT690 Steel for Predicting Deformation Process and Material Return by Numerical Simulation" includes:
- Analysis of the deformation behavior of TRIP HCT690 steel through material testing and tests considering stress states to define boundary conditions of the yield criterion and subsequent deformation behavior.
- Implementation of measured data in a numerical simulation environment to predict the deformation process and material return.
- Evaluation of the influence of the computational model and process parameters on the deformation process and material return.
- Comparison of numerical simulation results with experimentally produced sheet stamping to validate the simulation approach.
- Addressing the need for new technologies and methods to achieve the accuracy of shape and dimension of automotive body parts in response to evolving design changes and manufacturer requirements.
1. It would be beneficial to improve the presentation of Figures 15, 18, and 21. Perhaps splitting them into two figures for better readability by the reader.
2. Provide the correct number of significant figures and uncertainties for the parameters obtained in the fits and presented in the tables.
3. Remove the frames that are placed within the graphs and present the information throughout the text and result tables.
4. Limit the x and y axes of the graphs presented to the extent of the data/fits obtained. Such adjustment will enhance the visualization of the fit and data.
5. Transfer all information from "Figure 32. Definition of the yield criterion: Hill48 (left), Vegter Lite (middle), and Vegter Standard 599 (right) for material HCT690" to the text or tables.
6. In the comparison graphs of numerical simulation results, Is it possible to include a scale? This would facilitate visualization.
Overall, the manuscript is interesting and presents important results. After the revision, I recommend its publication.
Comments on the Quality of English LanguageModerate editing of English language required
Author Response
Dear Reviewer 4,
you can fing our response in the attachment.

Round 2
Reviewer 2 Report
Comments and Suggestions for Authors
The authors make all necessary modifications, and the manuscript is valid to be published in the current form.